# Transcriptome Profile Analysis of Winter Rapeseed (*Brassica napus* L.) in Response to Freezing Stress, Reveal Potentially Connected Events to Freezing Stress

**DOI:** 10.3390/ijms20112771

**Published:** 2019-06-05

**Authors:** Yuanyuan Pu, Lijun Liu, Junyan Wu, Yuhong Zhao, Jing Bai, Li Ma, Jinli Yue, Jiaojiao Jin, Zaoxia Niu, Yan Fang, Wancang Sun

**Affiliations:** 1College of Agronomy, Gansu Agricultural University, Lanzhou 730070, China; vampirepyy@126.com (Y.P.); 18894310220@163.com (Y.Z.); bj741912523@163.com (J.B.); 18189560623@163.com (L.M.); yuejinli1990@sina.com (J.Y.); jiao18894310228@163.com (J.J.); 15719327746@163.com (Z.N.); wujuny@gsau.edu.cn (J.W.); 2Gansu Provincial Key Laboratory of Aridland Crop Science, Lanzhou 730070, China; liulj198910@163.com (L.L.); ffyv@163.com (Y.F.)

**Keywords:** morphology, physiological, ultrastructure, transcriptomic, freezing stress

## Abstract

Winter rapeseed is not only an important oilseed crop, but also a winter cover crop in Northern China, where its production was severely limited by freezing stress. As an overwinter crop, the production is severely limited by freezing stress. Therefore, understanding the physiological and molecular mechanism of winter rapeseed (*Brassica napus* L.) in freezing stress responses becomes essential for the improvement and development of freezing-tolerant varieties of *Brassica napus*. In this study, morphological, physiological, ultrastructure and transcriptome changes in the *Brassica napus* line “2016TS(G)10” (freezing-tolerance line) that was exposed to –2 °C for 0 h, 1 h, 3 h and 24 h were characterized. The results showed that freezing stress caused seedling dehydration, and chloroplast dilation and degradation. The content of malondialdehyde (MDA), proline, soluble protein and soluble sugars were increased, as well as the relative electrolyte leakage (REL) which was significantly increased at frozen 24 h. Subsequently, RNA-seq analysis revealed a total of 98,672 UniGenes that were annotated in *Brassica napus* and 3905 UniGenes were identified as differentially expressed genes after being exposed to freezing stress. Among these genes, 2312 (59.21%) were up-regulated and 1593 (40.79%) were down-regulated. Most of these DEGs were significantly annotated in the carbohydrates and energy metabolism, signal transduction, amino acid metabolism and translation. Most of the up-regulated DEGs were especially enriched in plant hormone signal transduction, starch and sucrose metabolism pathways. Transcription factor enrichment analysis showed that the AP2/ERF, WRKY and MYB families were also significantly changed. Furthermore, 20 DEGs were selected to validate the transcriptome profiles via quantitative real-time PCR (qRT-PCR). In conclusion, the results provide an overall view of the dynamic changes in physiology and insights into the molecular regulation mechanisms of winter *Brassica napus* in response to freezing treatment, expanding our understanding on the complex molecular mechanism in plant response to freezing stress.

## 1. Introduction

Freezing stress (<0 °C) is one of the abiotic stresses, which limits the geographical distribution and seriously affects the growth, development, quality and productivity of crops. Global annual crop losses due to the low-temperature damage are up to hundreds of billions of dollars [1]. In January 2008, China suffered from a severe freeze injury, which had a serious impact on winter rapeseed production. The affected area accounts for about 77.8% of the country’s winter rapeseed area [2]. Therefore, it is important to study and excavate the freezing resistance related genes, breed freezing resistance varieties of winter *Brassica napus*, and develop the *Brassica napus* in northwest China.

Previous studies indicate that the freezing injured primary sites are the cell membrane systems, which causes cellular dehydration, reactive oxygen species accumulation, intercellular ice and protein denaturation, all of which are contributors to the membrane systems’ damage [3]. Under freezing stress, plants induce the genes which encode molecular chaperones, and synthesize specific proteins; these alterations protect plants and help them to withstand the freezing damage to the cell [3,4]. Research has shown that freezing tolerance is a quantitative trait that is governed by gene expression networks [3,5]. The mechanism of freezing stress in plants has been extensively studied for many years. The model plant *Arabidopsis thaliana* has yielded immense amounts of relevant research, helping to identify considerable amounts of freezing-induced genes, transcription factor families, signaling pathways and metabolism pathways [6].

High throughput sequencing is a high-efficiency and economically sounds analysis strategy. In order to investigate the transcription levels of a specific tissue at a specific time point, or to compare the differences in transcription levels among different genotypes/varieties, even in species that lack a reference genome, RNA sequencing (RNA-Seq) technology is a preferred strategy. RNA-Seq has been widely applied in investigating the global expression network in adversity stress in many species, such as *Vitis amurensis* [7], Taxus [8] and soybean [9], and helps to reveal the related metabolic pathways, such as taxol biosynthesis [8], anthocyanin biosynthesis [10] and regulatory events and related gene expressions. In recent years, plenty of research on *Brassica napus* have been made, including lignin biosynthesis [11,12], to identify candidate genes responsible for yield improvement [13], the genetic control of plant height [14], seed fatty acid biosynthesis [15], cadmium stress [16], water stress [17], drought stress [18,19] and cold stress [20,21,22,23], however, studies of freezing stress in *Brassica napus* are still lacking. Things such as freezing tolerance regulatory mechanisms underlining gene regulation, however, remain unclear in *Brassica napus*. Fortunately, the genome of *Brassica napus* has been sequenced [24,25,26], which provides a valuable molecular resource to investigate the stress tolerance in *Brassica napus*, and will help to understand the rapeseed molecular basis of freezing tolerance mechanisms in further studies.

In China, the production of winter *Brassica napus* is mainly distributed in the Huang-Huai River and Yangtze River basin [27], but the same cultured varieties cannot survive well through the winter in semi-arid regions of Northwest China. Additionally, due to the lack of suitable crop species to plant through winter, most cultivated land lies idle without any cover in winter and early spring in these regions. To solve these problems, in the past decade, we developed strong freezing tolerant winter *Brassica rapa* varieties (longyou7 [28,29,30]) and successfully adapted them in semi-arid regions of China such as Hexi corridor in Gansu province, Xinjiang province, Qinghai province, Ningxia Autonomous Region China [31]. It became an important oilseed crop and winter cover crop in north China. However, the winter *Brassica rapa* has some disadvantages, such as low seed oil quality and yield. Compared to this, *Brassica napus* has many advantages such as high yield, strong adaptability and excellent quality, and it has occupied a dominant position across the world. However, little is known about the complex regulatory mechanism of the winter *Brassica napus* response to freezing stress.

In this study, we selected the *Brassica napus* line “2016TS(G)10” which was developed by Gansu Agricultural University as a research material. 2016TS(G)10 was formed by hybridization of *Brassica rapa* (longyou7, a strong cold tolerance variety, bred by Gansu Agricultural University) as the female parent and *Brassica napus* (Vision, a winter ecotype, introduced from Europe) as the male parent. It has shown good winter survival ability and can survive over winter in Lanzhou of the Gansu province (China). Based on previous studies, before overwintering, the plants of “2016TS(G)10” showed winter turnip rapeseed growth habits, such as the top of the shoot being below the ground, the leaf color being dark green, the LT_50_ of field plant leaves (samples were harvested on 26 October) being −13.38 °C and the root cap being 0.59 and much bigger than the other lines (bred by Gansu Agricultural University, China). The overwintering rate for two consecutive years in Tianshui (Gansu, China) was 97.8%, and 25.5% (non-mulch)/~80.0% (plastic mulch) in Lanzhou (Gansu, China), so we selected “2016TS(G)10” as the strong cold tolerance material in further studies (article submitted in Scientia Agricultura Sinica, under review). Although the physiology, proteomics, and microRNA of cold tolerance of winter turnip rapes (*Brassica rapa* L.) have recently been studied, research of *Brassica napus* under freezing stress is rarely reported and the complex mechanism of freezing tolerance is still unclear.

In this present study, 50-day-old seedlings of freezing-tolerant *Brassica napus* line “2016TS(G)10” was treated at −2 °C for 0 h, 1 h, 3 h and 24 h. The frozen leaves were used to observe the phenotype, measure physiological indexes and analyze transcriptome data to investigate the molecular mechanism of freezing response. This research will expand our understanding on the complex molecular mechanism of freezing stress response in *Brassica napus*, and the transcriptome profiles analysis results provide candidate gene resources, which are important for improving and developing the freezing tolerance varieties. 

## 2. Results

### 2.1. Morphology and Physiochemical Changes of Winter Brassica napus after Low-Temperature Treatment

The seedlings were transferred to a low-temperature condition (4, 0, −2 and −4 °C) for 24 h (Figure 1A). There were morphological changes appearing in the seedlings under 4 °C treatments. Slight blade wilting was observed in the 0 °C treatment and it recovered at room temperature for 24 h. With the temperature decrease, the damage to the seedling increased. Freezing at −2 °C caused leaf and stalk dehydration, and the edges of leaves curled after 24 h; the survival rate was 52%. Upon being exposed to −4 °C, the blades and petioles were severely dehydrated and the survival rate was only 3.4% after recovery (Figure 1C). By comparing the morphological and survival rate under different low-temperature treatments (4, 0, −2 and −4 °C), we selected −2 °C as the freezing treatment condition in subsequent studies.

Upon exposure to −2 °C conditions, the seedlings were treated for 1 h and the older leaves observed a slight wilting, while the petioles were dehydrated and wilted after 3 h of freezing. After being continually treated for 24 h, the aboveground parts froze (Figure 1B). Previous studies generally used REL and MDA as direct stress markers to reflect the membrane damage by cold stress [32]. Changes of REL have shown that successive freezing stress caused irreversible damage to cells (Figure 1C). The MDA concentration exhibited a continuous increase and peaked at 24 h, with a 3.5 fold higher change than at 0 h (Figure 1C). The soluble sugar and proline content had significant changes; increasing rapidly and achieving a higher level after 3 h, then decreased slightly, while the soluble protein was significantly increased after 1 h of treatment (Figure 1C). This result showed that during the freezing periods, winter *Brassica napus* actively stimulated a series of physiological response mechanisms to resist freezing injuries. 

### 2.2. Ultrastructure Changes of Winter Brassica napus Leaves under Freezing Stress

In order to study the damage to chloroplast caused by freezing stress, we investigated the ultrastructure of leaves with a transmission electron microscope. The micrographs of chloroplasts indicated that starch grains significantly accumulated in chloroplasts. In the non-stress condition (0 h), chloroplasts were elongated and elliptically shaped, situated regularly along the cell wall, with a few starch grains (Figure 2—0 h). When stressed for 3 h, the chloroplasts were arranged sparsely and were swollen (Figure 2—3 h). After 24 h of freezing stress, the chloroplasts were approximately circular shaped, arranged irregularly in cells and scattered in the cytoplasm. Obvious gaps existed in the chloroplast grana and starch grains increased in quantity and volume (Figure 2—24 h).

### 2.3. Transcriptome Sequencing and Correlation Analysis

Given the physiological characteristics of *Brassica napus* seedlings under freezing stress, transcription level changes in the seedlings were analyzed using the Illumina HiSeq 2000 platform. Twelve RNA samples respectively harvested from non-frozen leaves (0 h) and frozen leaves that were exposed to −2 °C for 1 h, 3 h and 24 h, and prepared to construct four cDNA libraries for RNA-seq. As shown in Table 1, a total of approximately 87.49 GB of clean data were obtained from the cDNA libraries, the clean data of each sample reached 6.28 GB. The percentage of Q30 bases was greater than 92.48%, while the GC content of each sample was almost 48%. Tophat v2.0.12 was used to map the clean reads to the *Brassica napus* reference genome. As the results show, the percentage of clean reads that mapped into the reference genome ranged from 76.76 to 78.35%. About 72% of clean reads were uniquely mapped and used for subsequent analysis. Multiple mapped reads were excluded from the next analysis (Table 1 and Appendix A). The correlation of three biological replications at each time point was used to judge whether the samples were reliable. Based on the FPKM of each sample, we calculated the correlation values, which showed that the biological replications of non-freezing (0 h) were ≥0.96 and the freezing treated (1 h, 3 h and 24 h) were ≥0.971 (Appendix A). Correlation results indicated that the experimental samples and results were considered reliable for further analysis.

### 2.4. Defining Differentially Expressed Genes (DEGs) 

To analyze differences of the *Brassica napus* transcription levels during freezing stress periods, FPKM was used to calculate the transcripts of all the differentially expressed genes. The fold change was calculated based on a comparison of the FPKM between the frozen-stressed and non-stressed sample. When the fold change of gene expression level was at least a two-fold change and Chi-square test (*p* < 0.05) FDR < 0.01, it was considered as a differential expression gene (DEG). A total of 690, 2538 and 2403 DEGs were identified in the groups of 1 h/0 h, 3 h/0 h, and 24 h/0 h, respectively. The bar charts reflect the number of up- and down-regulated DEGs in the three groups under freezing stress. After 1 h of freezing stress, the numbers of down-regulation DEGs (407) were more than those of up-regulation (283). With the freezing stress time increasing, the number of up-regulated DEGs increased from 1490 (3 h) to 1528 (24 h), and the down-regulated DEGs decreased from 1048 (3 h) to 875 (24 h) (Figure 3A and Appendix A). Comparing the three groups to yield a total of 3905 DEGs, the Venn diagram can reflect the number of up-regulated and down-regulated DEGs, as well as specific and commonly regulated DEGs in different groups. Among them, 147 (67 up-regulated and 80 down-regulated), 1246 (691 up-regulated and 555 down-regulated) and 1138 (704 up-regulated and 434 down-regulated) DEGs belonged to 1 h, 3 h and 24 h, respectively. The 349 DEGs (133 up-regulated and 216 down-regulated) were commonly regulated by freezing stress in the three groups (Figure 3B and Appendix A). To more intuitively show the differences and similarities among the freezing-responsive DEGs in different stress time in *Brassica napus* leaves, hierarchical clustering was developed to represent the expression of 3905 DEGs (Figure 3C). The results showed that there was a significant difference in gene expression profile differences among the four time points. As the stress time increased, the genes with high expression at 0 h were gradually down-regulated, while those with low FPKM at 0 h were gradually up-expressed. This result indicated that freezing treatment suppresses a set of gene expression or induces another set of genes expression, though these changes were to regulate the freezing tolerance. To be clear about the distribution of DEGs in subgenomes C and subgenomes A, we had drawn a Circos plot to show the distribution of DEGs on 19 chromosomes. A total of 3109 DEGs were accurately positioning on 19 chromosomes, 1690 DEGs on subgenomes C and 1419 on subgenomes A. The detailed list was shown in Figure 3D and Appendix A.

### 2.5. Expression Pattern and Functional Analysis of the DEGs in Winter Brassica napus

We performed the functional annotation of unigenes by BLAST against eight databases, such as Nr, Swiss-Prot, GO, COG, KOG, KEGG, eggNOG, and Pfam (Appendix A). According to the expression profiles, 3905 DEGs were classified into 6 clusters by co-expression clustering (Figure 4A and Appendix A). A total of 876 genes were classified into cluster 1, which rapidly down-regulated during 0 h to 3 h, then gradually up-regulated at 24 h. This result indicated that those genes were transiently repressed by freezing. Most of them participated in pathways such as “plant hormone signal transduction”, “starch and sucrose metabolism” and “biosynthesis of amino acids”. Genes belonging to cluster 4 were persistently down-regulated; the transcriptions were suppressed by freezing stress, the majority of them enriched in the “ribosome” and “ribosome biogenesis in eukaryotes” pathways. Cluster 5 contained fewer genes (189), the expression level of these genes were significantly up-regulated during freezing stress, most of them enriched in “plant hormone signal transduction” and “plant-pathogen interaction” pathways. Compared with cluster 5, the expression of genes in cluster 2 was also up-regulated but relatively lower. Cluster 6 had nearly a third of the DEGs (1417) that participated in KEGG pathways (Figure 4B and Appendix A).

#### 2.5.1. GO Classification Analyses 

All unigenes were analyzed in the GO database. Annotated genes were divided into three major functional categories: biological processes (BP), cellular components (CC) and molecular functions (MF). Most transcripts are enriched in the cellular process (GO: 0009987), single-organism process (GO: 0044699), metabolic processes (GO: 0008152), response to a stimulus (GO: 0050896) and biological regulation (GO: 0065007) in the BP category. Enrichment also occurred for genes associated with CC, such as the cell (GO: 0005623), cell part (GO: 0044464), organelle (GO: 0043226) and membrane (GO: 0016020). Binding (GO: 0005488) and catalytic activity (GO: 0003824) in MF were highly enriched in both up- and down-regulated DEGs. Compared with down-regulated transcripts, the cell killing (GO: 0001906) and extracellular region part (GO: 0044421) terms were peculiar in up-regulated DEGs; the locomotion (GO: 0040011) and biological phase (GO: 0044848) were unique in upregulation GO terms (Figure 5A,B and Appendix A). It is obvious that there were more functional terms for biological processes and relatively few transcripts for cellular component and molecular function.

#### 2.5.2. KEGG Annotation Analyses 

KEGG pathway analysis was conducted to investigate whether the freezing stress-responsive genes in *Brassica napus* were involved in some special pathways. A total of 887 unigenes were mapped to 108 KEGG pathways. Metabolic pathways were enriched in 482 DEGs (Appendix A). Top20 of KEGG enrichment showed that up-regulated DEGs were highlighted in “plant hormone signal transduction (ko04075)”, “Phenylalanine metabolism (ko00360)”, “pentose and glucuronate interconversions (ko00040)”, “phenylpropanoid biosynthesis (ko00940)”, “starch and sucrose metabolism (ko00500)” and “plant-pathogen interaction (ko04626)” (Figure 5C, Appendix A and Appendix A). Moreover, “ribosome (ko03010)”, “plant hormone signal transduction (ko04075)”, “biosynthesis of amino acids (ko01230)”, “ribosome biogenesis in eukaryotes (ko03008)” and “protein processing in endoplasmic reticulum (ko04141)” annotated the most down-regulated DEGs (Figure 5D, and Appendix A and Appendix A). The level of transcripts related to signal transduction was altered in response to freezing (Appendix A). For example, BnaC03g39170D encoding an auxin-responsive protein IAA19, was notably continuously up-regulated in the freezing treatment. In addition, it is noteworthy that the gene of BnaA03g42940D that is enriched in “starch and sucrose metabolism (ko00500)” pathway, encode glycosyl hydrolase families 14, and the expression also significantly increased by freezing stress (Appendix A). It is obvious that there were more functional terms for biological processes and relatively few transcripts for cellular components and molecular functions.

### 2.6. Changes of Differential Expression Transcription Factors (TFs) under Freezing Stress

Among the 3905 DEGs, around 282 up-regulated DEGs and 115 down-regulated ones were identified as transcription factors (TFs) under continuous freezing stress. All of them fell into 45 TF families (Figure 6A and Appendix A). Most of the up-regulated TFs belonged to the AP2/ERF (55) (including AP2/ERF-AP2, AP2/ERF-ERF and AP2/ERF-RAV), MYB (38), and WRKY (31) families. The down-regulated TFs mostly belonged to bHLH (12), MYB (11), C2C2 (11) and also some AP2/ERF families (10) (Figure 6A). With respect to the specific and common TFs, the number of up-regulated specific TFs was much larger (153) than the down-regulated ones (84), the common up-regulated TFs (24) were 2.4 times larger than down-regulated (11) (Figure 6B). The different expression TFs, including 1 h/0 h, 3 h/0 h and 24 h/0 h, were identified in our analysis and listed in Appendix A.

### 2.7. RNA-Seq Expression Validation by qRT-PCR

The quantitative real time-PCR (qRT-PCR) technology was used to determine the reliability of our transcriptome data. Based on high FPKM and fold change, we selected 20 DEGs as targets that were closely associated with the freezing response. The result confirmed that qRT-PCR expression patterns were in good agreement with the RNA-Seq trend (*R*^2^ = 0.92) (Figure 7). This indicated that the RNA-Seq results were reliable in the present study.

## 3. Discussion

Winter rapeseed is not only an important oilseed crop, but also an important winter cover crop in the north of China [33]. Freezing damage is one of the main environmental factors limiting the production of winter rapeseed. To improve the freezing tolerance, it is necessary to investigate the freezing tolerance mechanism of winter *Brassica napus*. Although the adaptation [34], physiology [35], proteomics [36], and transcriptome [21] of *Brassica napus* under cold stress have been studied in recent years, freezing tolerance research has rarely been reported, the complex mechanism of freezing tolerance in *Brassica napus* is still limited. Therefore, to elucidate the responses of freezing tolerance in winter *Brassica napus*, physiological and transcriptome analyses were performed in this research.

### 3.1. Freezing Stress Affects Physiological Changes in Seedlings of Brassica napus 

Freezing stress induces a state change from water to ice, accompanied by the occurrence of hyperosmotic and mechanical stresses, disrupting the plasma membrane, directly affecting the plant’s survival [22,23]. As previously reported, electrolyte leakage and MDA can reflect the degree of membrane damage in stress [32,37]. Additionally, the accumulation of proline, soluble protein and soluble sugars, facilitate osmotic potential and plays an important metabolic role in stress tolerance [38]. Proline and soluble protein participated in stabilizing protein synthesis, protecting enzyme activity and regulating cytoplasmic acid and alkalinity [39,40]. As a binding substance of membrane lipids, soluble sugar kept the stability of membranes [41]. As previously studied, the contents of MDA, proline and soluble sugars were increasing, as well, electrolyte leakage has been illuminated in cold-tolerant species, such as *Santalum album* L. leaves [42], *Vitis amurensis* [7] and tomato [43]. Our results were consistent with those reports. Over the 24 h time course of freezing stress, seedling dehydration, chloroplast dilation and degradation, an increase in the content of MDA, proline, soluble protein and soluble sugars, as well as REL showed rapid growth at the frozen 24 h (Figure 1C). The increase of osmotic adjustment substance content could reduce the freezing point to increase plant cell membrane stabilization and protect membrane integrity during freezing-caused dehydration in *Brassica napus*. Prolonged freezing stress triggers intracellular and extracellular freezing, which will cause mechanical damage to cells and lead to plant death. Taken together, these physiological changes indicate that the *Brassica napus* line “2016TS(G)10” has developed the ability to withstand short-period freezing stress.

### 3.2. Plant Hormone Signal Transduction Related Freezing Stress

Phytohormones are small endogenous signaling molecules, such as gibberellin (GA), auxin (IAA), cytokinin (CK), brassinosteroid (BR), abscisic acid (ABA), ethylene (ET), jasmonic acid (JA), salicylic acid (SA), and strigolactone (SL), which orchestrate a dual function. Plant hormones are mediators that not only act on endogenous developmental processes, but also convey environmental stimuli to multiple hormone-response pathways for adaptation to adverse situations [44,45,46]. Jain and Khurana [47] found rice auxin-responsive related genes are differentially regulated during reproductive development and abiotic stress. Similar results were also obtained in the present study. There were 118 genes enriched in “plant hormone signal transduction” (Appendix A), 44 DEGs were up-regulated and belonged to auxin signaling, encoding “transmembrane amino acid transporter protein”, “auxin responsive protein”, “AUX/IAA family” and “GH3 auxin-responsive promoter” (Appendix A). The *Aux/IAA* genes are key regulators of auxin-modulated gene expression that are themselves auxin-inducible [48]. In Arabidopsis and Rice, a significant number of auxin-regulated genes are additionally affected by cold stress [44,49]. When auxin analogs were applied on *Brassica napus*, it stimulated the accumulation of freeze-protective metabolites and soluble sugars during cold hardening [50]. Therefore, we speculated that auxin might play an important role in *Brassica napus* response to freezing stress. Genes involved in ABA, BR, GA, ET, and JA regulation were also identified in our study. A total of 19 DEGs encoded three key proteins, which were, respectively, the “Abscisic acid receptor” (1 gene up-regulation), “Protein phosphatase 2C” (13 genes down-regulation) and “ABSCISIC ACID-INSENSITIVE 5-like protein” (1 genes up-regulation and 4 genes down-regulation), which were involved in ABA signaling pathway. This indicated that the freezing-induced expression of related genes, like as BnaC07g34880D, BnaAnng26550D, in *Brassica napus* appeared to be ABA-dependent.

### 3.3. Ribosome-Related DEGs and Amino Acid Metabolic Pathways under Freezing Stress

The ribosome is the critical site for protein synthesis, which is integral to the translation of mRNA into proteins, composed of small and large subunits [51]. In this present study, a large number of down-regulated DEGs were enriched in “Ribosome (ko03010)” and “Ribosome biogenesis in eukaryotes (ko03008)” (Appendix A). In the initial phase (24 h) of deacclimation in *Arabidopsis thaliana*, most of the down-regulated transcripts were associated with ribosome biogenesis related genes [52]. Zeng et al. found that a large number of up-accumulated proteins were involved in the “Ribosome” after freezing treatment of the pre-CA (cold acclimation) plants [28]. Additionally, the same result was demonstrated in the proteomics of *Flammulina velutipes* mycelia cold resistance [53]. Ribosome biogenesis, a cellular process producing ribosomes, is essential for cell growth and cell proliferation [54]. It can be assumed that the *Brassica napus* seedlings were severely damaged and Ribosome biogenesis was repressed under freezing stress, the degradation of ribosomes already present in the cells and a remobilization of the resulting amino acids for freezing tolerance-related protein biosynthesis to respond to the initial freezing stress.

Amino acids’ metabolic pathway being was significantly responsive to cold/freezing stress has been reported in *Arabidopsis* [55], mycelia [53], loquat [56], ryegrass [55] and peach [57] under cold/freezing. Our results showed that several amino acid metabolic pathways were enriched after freezing stress, for a total of 121 DEGs, such as “cysteine and methionine metabolism” (28 DEGs), “arginine and proline metabolism” (26 DEGs), “glycine, serine and threonine metabolism” (14 DEGs), “phenylalanine metabolism” (39 DEGs), “alanine, aspartate and glutamate metabolism” (12 DEGs), “tryptophan metabolism” (17 DEGs), “tyrosine metabolism” (16 DEGs), “lysine biosynthesis” (6 DEGs), “phenylalanine, tyrosine and tryptophan biosynthesis” (11 DEGs s), “histidine metabolism” (6 DEGs), and “valine, leucine and isoleucine biosynthesis” (1 DEG) being annotated (Appendix A). Simultaneously, the content of proline was markedly increased and achieved a higher level after freezing for 3 h (Figure 1C). Proline is well documented to be associated with plant abiotic tolerance in multiple ways, as a compatible osmolyte, molecular chaperone, and ROS scavenger [58]. Several comprehensive studies showed that the proline metabolism has a complex effect on development and stress responses and that proline accumulation is important for the tolerance of certain adverse environmental conditions [59,60,61]. Proline levels are usually determined by the balance between the biosynthesis and catabolic pathways, the biosynthesis pathways were controlled by *P5CS* genes [62]. In this study, the expression profile of the “arginine and proline metabolism (ko00330)” DEGs were analyzed and most of these DEGs were up-regulated in response to freezing stress. It is shown from the data analysis that the expression of 5 genes related with *P5CS* (BnaC04g05620D, BnaA05g05760D, BnaA09g35230D BnaA04g22810D and BnaC04g46630D) were down-regulated, but the FPKM of 2 genes (BnaC04g05620D and BnaA05g05760D) were still the highest among the DEGs involved in the “arginine and proline metabolism” pathway. Proline catabolism occurs in mitochondria via the sequential action of proline dehydrogenase or proline oxidase (*PDH* or *POX*) producing *P5C* from proline and *P5C* dehydrogenase (*P5CDH*). *P5CDH* has been found in chloroplasts by proteome analysis, suggesting that *P5C* is also converted to glutamate in plastids [63]. In the present study, BnaA06g39660D, BnaAnng07910D and BnaC02g38230D coded proline dehydrogenase, were up-regulated, but the FPKM of these were the lowest among the DEGs involved in this process. The above results show that “Ribosome”, “Ribosome biogenesis”, amino acid-related metabolic and proline biosynthesis and catabolic might play some essential roles in response to the freezing stress of *Brassica napus*.

### 3.4. Starch and Sucrose Metabolism Pathway

The carbohydrate and energy metabolism is the basis for the maintenance of cell life [64], and the carbon source mainly arises from the degradation of starch; the enzymes or genes involved in starch degradation are crucial for sugar metabolism. Our transcriptome data showed that a total of 208 DEGs were enriched in carbohydrate and energy metabolism pathways (Appendix A). Especially the “starch and sucrose metabolism” and “pentose and glucuronate interconversions” were significantly enriched, which reflects that the soluble sugar content increased under the freezing treatment (Figure 1C). Previous studies have clearly demonstrated that there was a strong correlation between the sugar concentration and cold tolerance [65], such as higher levels of soluble sugars exhibited in eskimo1 (*esk1*) locus mutant plant, which was frost tolerant [66]. In overexpressed *CBF3* in cold tolerant Arabidopsis, the sugar levels of transgenic plants were higher than the control [67].

Sugars have several beneficial effects in protecting plants against chilling, including osmoregulation, sugar signaling, sugar-modulated gene expression, support of anti-oxidative metabolism, and direct antioxidant activity against reactive oxygen species [68]. *β-amylase* (*BAM*) is implicated in the synthesis or degradation of soluble sugars [69,70]. Because of its unique role in connecting starch breakdown and the downstream sugars at various forms, *BAM* is considered to play a specific role in the accumulation of soluble sugar under cold stress. In this study, *BAM1* (BnaC09g21440D and BnaA07g05790D) and *BAM3* (BnaC07g34180D, BnaA03g42940D and BnaA01g17940D) were identified to be up-regulated, with the expression having increased rapidly after freezing treated for 1 h and 3 h, and the expression was slightly up-regulated after freezing for 24 h (Appendix A). Simultaneously, the determination of soluble sugar content showed the same trend (Figure 1C). Another research article indicates that the overexpression of *PtrBAM1*(*β-Amylase*) in tobacco (*Nicotiana nudicaulis*) increased the BAM activity, promoted starch degradation and enhanced the contents of maltose and soluble sugars, whereas opposite changes were observed when the *PtrBAM1* homolog in lemon (*Citrus lemon*) was knocked down [71]. Silencing of *AtBAM3* resulted in the reduction of soluble sugars [72,73]. In another work, the deletion of an *AtBAM3* orthologue in potatoes caused starch accumulation [73,74]. *PtrBAM1* is localized in the chloroplast, where starch is broken down to produce maltose [71]. As the initial attack on the starch grains is mainly catalyzed by BAM [75], thus, it is suggested that the expression of *BAM1* and *BAM3* were repressed under continued freezing stress, the catalytic of BAM reduced, and the starch grains significantly accumulated after frozen 24 h (Figure 2). In this study, *BAM1* (BnaC09g21440D) and *BAM3* (BnaC07g34180D and BnaA03g42940D) have high FPKM and log2 FC. It is suggested that *BAM* play an importance role in the freezing tolerance of *Brassica napus*.

Furthermore, sugars are involved in modulating the expression of stress-related genes in multiple stress responses, which are on their turn tightly regulated by the circadian clock [76]. After 24 h of frozen stress, in the group of down-regulated DEGs, the “circadian rhythm-plant (ko04712)” was identified (Appendix A). According to the results of previous studies, *TOC1*, members of a small family of proteins, designated as Arabidopsis *PSEUDO-RESPONSE REGULATORS* (*APRR1/TOC1*, *APRR3*, *APRR5*, *APRR7*, and *APRR9*), is believed to be a component of the central oscillator in *Arabidopsis thaliana* [77]. In our study, the *APRR7* (BnaCnng03230D), *APRR5* (BnaC07g29960D, BnaA06g26980D and BnaC07g29960D) were identified, but, it was not yet certain whether or not other PRR family members (*PRR9*, *PRR7*, *PRR5* and *PRR3*) were implicated in clock function per se [78]. 

### 3.5. Lipid Metabolism 

Membrane systems are the primary site of freezing injury in plants and are usually damaged by freeze-induced cellular dehydration [79]. In the present study, the results of electrolyte leakage and MDA increased under freezing (Figure 1C), which demonstrated that the membranes were clearly destroyed. When plants come to extreme environments, such as cold stress, they have to adapt the membrane composition in order to maintain organelle function [80]. Correspondingly, we identified a total of 90 DEGs involved in “lipid metabolism”. A total of 54 out of these DEGs involved in this process were up-regulated and 39 DEGs were down-regulated. Previous studies have shown that the cytomembrane becomes less fluid under cold stress, increasing the degree of unsaturation in the membrane lipids so as to maintain the normal fluidity of membranes [80,81], and high unsaturated fatty acid levels can improve the cold tolerance and prevent cell damage [82]. In our study, 3 genes (BnaC06g28830D, BnaC04g40760D and BnaC05g37450D) that were involved in “Biosynthesis of unsaturated fatty acids”, which contributed to the freeze tolerance by altering the lipid composition, were significant up-regulation (Appendix A). This indicated that the high expression levels of these genes may be an important functional trait in the freeze tolerance mechanism found in *Brassica napus*. In addition, based on the above results, the accumulation of soluble sucrose and proline also play an important role in maintaining membrane stability (Figure 1C). So, the membrane protection system network was very complicated.

### 3.6. Transcription Factors (TFs) Induced in Freezing Stress

Transcription factors, via specific binding to the cis-acting element in their promoters, are very important in mediating the signaling pathway in response to freezing stress. In the present study, we identified 397 TFs differentially expressed in *Brassica napus* under freezing stress. Among these TFs, the AP2/ERF (60) family contains most members (Figure 6), implying their critical roles in the *Brassica napus* response to freezing conditions. AP2/ERFs, a large transcription factors family in plants, includes four major subfamilies: AP2, RAV, ERF and DREB [83]. The subfamilies of *DREB* and *ERF* are especially viable candidates for crop improvement, because of the tolerances to drought, salt, freezing and multiple diseases, which were enhanced in the overexpression transgenic plants [84]. In our study, no *DREB* was identified and a larger member in the ERF subfamily was identified, which is not only considered to be a mediator of ethylene-related responses, but also plays an important role in plant tolerance to drought, salt, freezing and diseases [84,85]. Leaves of *Brassica napus* exposed to cold stress (4 °C) for 24 h, the RAV and DREB subfamilies were expressed at the early stage, the AP2 subfamily was expressed later [21]. Previous RNA-seq results have confirmed that most of the AP2 and ERF members were up-regulated in loquat fruitlets under freezing stress (−3 °C) [56]. Zheng et al. [85] suggested that DREB/CBF TFs play a vital role in the chilling, but not the freezing stress response in tea. These results indicated that different subfamilies respond to cold and freezing differently, and different subfamilies were also involved in different response stages. We did not identify the DREB and CBF families in our study, but a very significant AP2-ERF family was identified, indicating that AP2-ERF plays an important role in freezing tolerance of *Brassica napus*. Other common TF families (Figure 6) involved in plant abiotic stress process, such as WRKY, NAC, bHLH etc., were also identified in our transcriptomes, which consist with previous reports in other plants.

Although the *CBFs* play important roles in cold acclimation, less than 20% of *COR* genes are regulated by the *CBFs*, suggesting the involvement of *CBF*-independent transcription factors in the regulation of their expression [86,87,88,89]. It is noteworthy, in the present study, in addition to up-regulated genes of AP2 family with high FPKM, C2H2 (*ZAT10* and *ZAT12*) and HSF (*HSFC1*, Heat stress transcription factor C-1) were also highly expressed (Appendix A). *ZAT12* and *ZAT10* belong to the cysteine-2/histidine-2 (C2H2)-type zinc finger protein family found in *Arabidopsis* [86,90,91]; *HSFC1* is a member of the Heat stress transcription factor family [92]. In this regard, previous studies have shown that a microarray analysis to identify five transcription factor genes, *HSFC1* (*Heat shock transcription factor c1*), *ZAT12* (*Zing finger of Arabidopsis thaliana 12*), *ZF* (*Zinc finger*), *SZF2* (*Salt-inducible zinc finger 2*) and *ZAT10 (Zing finger of Arabidopsis thaliana 10)*, that are co-expressed in parallel with *CBF2*, and which positively regulate *COR* gene expression and freezing tolerance [86,93]. Park et al. identified 27 genes in Arabidopsis Col-0 that encoded transcription factors and were rapidly induced in response to low temperatures in parallel with *CBF1*, *CBF2* and *CBF3* [86]. Further, they established that five of these “first-wave” genes, *HSFC1*, *ZAT12*, *ZAT10*, *ZF*, and *CZF*, encoded transcription factors that were able to induce one or more of 35 *CBF* regulon genes. The results were confirmed in another study. Park S et al. found that *HSFC1*, *ZAT12*, *ZAT10*, *ZF*, *CZF* were also rapidly cold-induced in two Arabidopsis ecotypes plants (Italy and Sweden), indicating that they might also have a role in co-regulating *CBF* regulon genes in different ecotypes [93]. Overexpression of *ZAT12* and *HSFA1*, which are rapidly induced in parallel with the *CBF* genes, has been shown to cause an increase in Arabidopsis freezing tolerance [86,91]. As alluded to the above, our results indicated that the *ZAT12*, *ZAT10* and *HSFC1* potentially contributed to the freezing tolerance in 2016TS(G)10 of *Brassica napus*, and could be the targets for further functional characterization.

## 4. Materials and Methods

### 4.1. Plants Growth 

The winter *Brassica napus* line “2016TS(G)10”, bred by Gansu Agricultural University (Gansu, China), was used in this experiment. Seeds were germinated in a culture dish on filter papers wetted by deionized water and maintained at 22 ± 1 °C until showing the cotyledon. Seedlings were transferred to pots filled with a 3:1 mixture of nutritional soil and vermiculite, and are grown in an illumination incubator with normal conditions (22/20 °C, day/night temperature, 16 h photoperiod, intensity of illumination 6000 Lx). The seedlings grew in uniform status for 50 days, and then were transferred into a pre-cooling incubator for freezing treatment.

### 4.2. Cold Treatment and Morphology Observation 

Cold treatments were set for 4, 0, −2 and −4 °C, respectively treated with 24 h (a 16 h photoperiod and intensity of illumination 6000 Lx). At each temperature, we treated 20 seedlings, repeated three times. After the low-temperature treatment, the plant was transferred to room temperature for recovery 2 weeks, and then the survival rate was measured.

Seedlings under −2 °C were treated for 24 h (intensity of illumination 6000 Lx, 16 h photoperiod). To avoid changes caused by the circadian rhythm, freezing stress treatments were started at 8 AM under light, from which non-treated samples (0 h) were obtained, and continued for 1, 3 and 24 h (8 AM of the next day), respectively, to obtain frozen-treated samples. At each time point, the second leaf that was count from the inside were harvested and frozen in liquid nitrogen immediately, then stored at −80 °C freezer for further study. Each time point had three biological replicates.

### 4.3. Physiological Measurement 

After the freezing treatment, we immediately measured the relative electrolyte leakage with fresh materials. The content of soluble sugar, soluble protein, malondialdehyde (MDA) and free proline measured using the frozen sample. Relative electrolyte leakage (REL) was measured by a digital conductometer DDS11A (Leici Instrument Factory, Shanghai, China) according to Bajji, Kinet et al [94]. Soluble sugar content was measured as described by Buysse and Merckx [95]. Soluble protein content was measured using coomassie brilliant blue staining [96]. MDA was determined by the thiobarbituric acid (TBA) reaction [97], and the free proline content was measured by the sulfosalicylic acid-acid ninhydrin method [98]. The SPSS statistical software package (SPSS version 22.0 Inc., USA) was used for the significant differences analysis.

### 4.4. Cell Ultrastructure Observations

After freezing treatment, leaf specimens (cut into 2 × 3 mm) were immersed in 2.5% (*v*/*v*) glutaraldehyde (8% (*w*/*v*) saccharose in 0.1 M pH 7.4 PBS) and fixed for 24 h at 4 °C. They were then washed 3 times with the same buffer, post-fixed with 1% osmium tetroxide at 4 °C for 5 h, then washed with 0.1 M sodium phosphate buffer for 20 min. Subsequently, the fragments were dehydrated with 50, 70, 80, 90 and 100% acetone, and soaked in a mixture of acetone: resin (1:1) at room temperature for 5 h, then, fixed in 100% resin overnight, and embedded in Spurr’s medium. The ultrathin sections were picked up on copper grids, stained with 2% uranyl acetate and Reynolds’ lead citrate (20) for 7 min and observed under the TEM (Hitachi H-7650, Tokyo, Japan).

### 4.5. Library Construction, Illumina Sequencing and Data Analysis

The total RNA of the aforementioned samples was used to construct an RNA-Seq library, which was completed by Biomarker technologies (Beijing, China). The RNA concentration was measured using NanoDrop 2000 (Thermo) and RNA integrity was assessed using the RNA Nano 6000 Assay Kit of the Agilent Bioanalyzer 2100 system (Agilent Technologies, CA, USA). The mRNA was purified from the total RNA using poly-T oligo-attached magnetic beads and interrupted randomly into short fragments. The short fragments were used as the template to synthesize cDNA, and the library was built. Illumina HiSeq 2000 sequencing was performed and paired-end reads were generated. The sequenced raw reads have been submitted to the SRA at NCBI with the accession numbers: SRP540905. After RNA sequencing, clean reads were obtained by removing containing adapter reads, ploy-N reads and low quality reads from raw reads. All clean data were calculated by Q20, Q30, GC-content and sequence duplication level, and then mapped to the reference genome sequence by Tophat2 [99] tools soft, to obtain the position information on the reference genome or gene, and the sequence characteristic information of the sequencing sample. Only reads with a perfect match or one mismatch were further analyzed and annotated.

#### 4.5.1. Differential Expression Genes (DEGs) Analysis

Identification of differential genes used FPKM (fragments per kilobase per million reads) [100]. Statistical comparison of FPKM values between freezing-stressed and non-stressed samples was performed using a web tool IDEG6 [101]. False discovery rate (FDR) <0.01 and log2 Fold Change ≥2 based on three biological replicates were considered as DEGs.

According to the information resource database of *B. napus* (http://www.genoscope.cns.fr.), the positional information of all DEGs was investigated. The positions of DEGs on chromosomes were drawn using Circos v0.69 [102].

#### 4.5.2. Functional Analysis of DEGs 

The functional of DEGs were identified through GO, COG, KEGG, KOG, NR, Pfam, Swiss-Prot and eggNOG database. GO was implemented by the GOseq R package, in which gene length bias was corrected, and the *p* value of DEGs ≤ 0.05 was considered as significantly enriched. KOBAS software was used to test the enrichment of DEGs in the KEGG pathway. Pathways with a *p* value ≤ 0.05 were defined as significant levels of differential expression.

### 4.6. Validation of DEGs by qRT-PCR 

A total of 2 μg templates RNA extracted from 1.2 was used to synthesize the first strand of cDNA in a 20-μL reaction using oligo dT primers, according to the manufacturer’s instructions of Takara Prime Script TM RT Master Mix (Perfect Real Time) kit. The expression patterns of 20 DEGs identified by RNA-Seq in this study were validated by q RT-PCR, performed using Takara SYBR Premix Ex Taq™ II kit (Takara) and run on the CFX Manager thermal cycler (Bio-Rad, CA, USA). Data of each sample were calculated in relation to the reference genes *F-box* and *SAND* using the 2^−ΔΔ*C*T^ method [103]. All of the treatments were tested for four technical replicates and three biological replicates. The gene names and specific primer sequences were detailed in Appendix A.

## 5. Conclusions

In conclusion, through physiological tests and RNA-seq technology, we strived to provide an overall view of the dynamic changes in physiology and insights into the molecular mechanisms of winter *Brassica napus* line “2016TS(G)10” in response to continuously freezing stress. Freezing stress caused seedling dehydration, chloroplast dilation and degradation, increased the content of MDA, proline, soluble protein and soluble sugars, and caused REL to rapidly grow at frozen 24 h. A total of 98,672 unigenes were annotated in *Brassica napus* and 3905 unigenes were identified as differentially expressed genes (2312 (59.21%) up-regulated and 1593 (40.79%) down-regulated). Annotating the differentially expressed unigenes into KEGG database, a series of freezing-resistant events were enriched, such as “plant hormones signal transduction (ko04075)”, “ribosomal (ko03010)”, “ribosome biogenesis in eukaryotes (ko03008)”, “biosynthesis of amino acids (ko01230)” and “starch and sucrose metabolism (ko00500)”. AP2/ERF, MYB, WRKY transcription factor families were mainly identified. A graph of physiological and transcriptome analysis in winter *Brassica napus* under course-time freezing stress is shown in Figure 8 and Appendix A. Although this study cannot completely illuminate the freezing tolerance mechanism about this line of winter *Brassica napus*, it serves as a molecular basis to explore candidate events in further studies. Therefore, our study is definitely applicable, as it not only expands the understanding of the molecular mechanism in freezing stress response, but also serves as a candidate gene resource for rapeseed for freezing tolerance breeding.

## Figures and Tables

**Figure 1 ijms-20-02771-f001:**
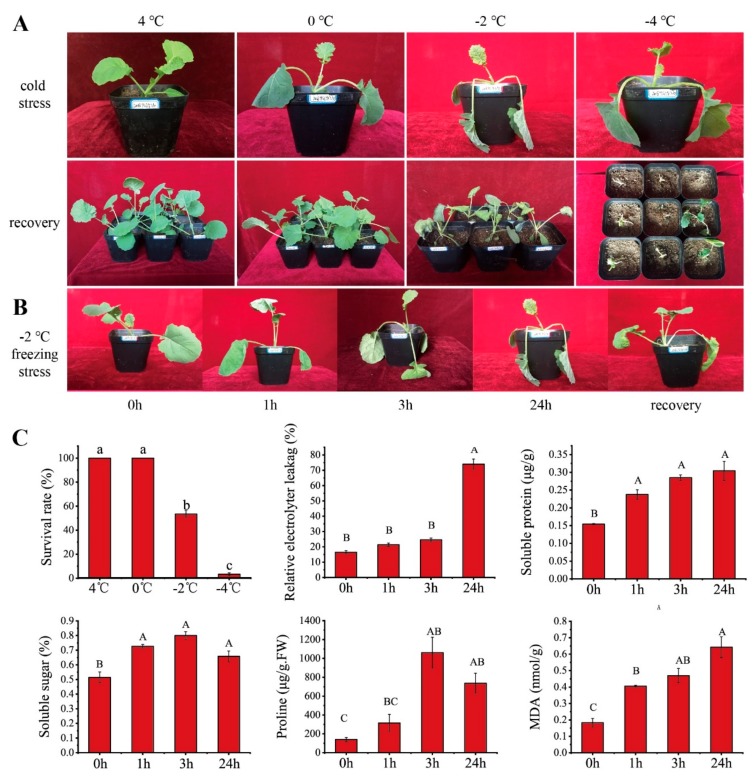
The morphology and physiochemical changes of winter type *Brassica napus* in cold-stress. (**A**) The seedlings were cold stressed for 24 h in 4, 0, −2, −4 °C and recovered after 24 h. (**B**) The morphology changes of seedling successive freezing for 24 h. (**C**) The survival rate, Relative electrolyte leakage, soluble protein, soluble sugar, Proline and MDA content accumulation in leaves of winter *Brassica napus* stressed with −2 °C for 0, 1, 3 and 24 h. The majuscules indicate a significant difference (*p* < 0.01) for the data of the stress-treated samples compared with unstressed samples. The mean values were calculated from three biological replicates. Error bars denote standard error of the mean.

**Figure 2 ijms-20-02771-f002:**
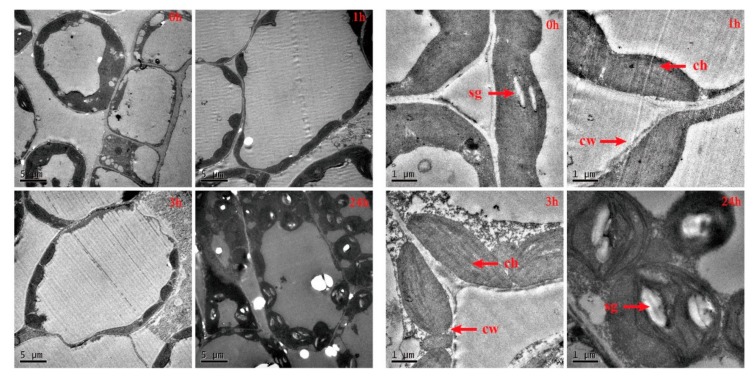
The transmission electron micrographs of chloroplast in the mesophyll cell which winter *Brassica napus* was exposed to in −2 °C with a continuous treatment (0, 1, 3, 24 h). The figure on the right magnification is ×4000; the figure on the right magnification is ×20,000. ch: chloroplast, sg: starch grain, M: mitochondria, CW: cell wall.

**Figure 3 ijms-20-02771-f003:**
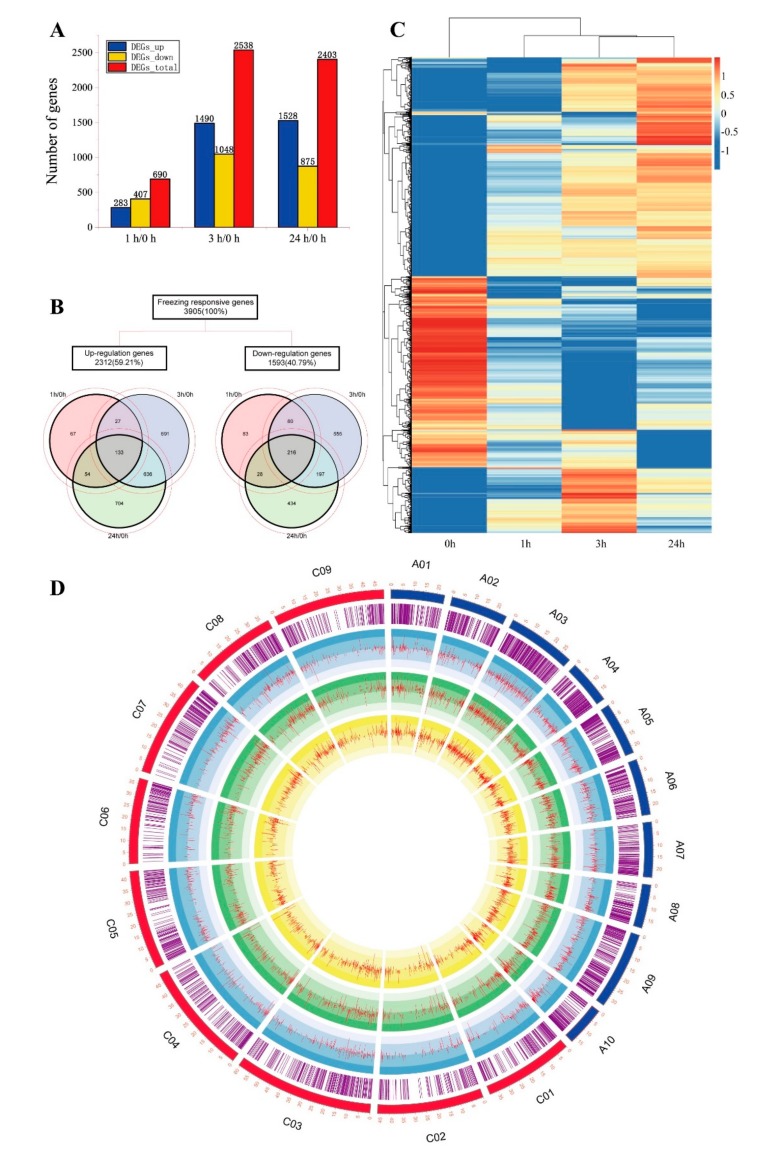
The expression profile of chilling regulated differentially expressed genes (DEGs) in *Brassica napus* leaves. (**A**) Column diagram representing the numbers of DEGs in three groups. (**B**) Venn diagrams representing the numbers of DEGs and the overlaps of sets obtained across three comparisons. (**C**) The heat maps representing 3905 DEGs expression profiles after freezing treatment. (**D**) the Circos Plot shows that the distribution of DEGs on 19 chromosomes and expression of DEGs in different time points. Red and blue showed the sizes of the 19 chromosomes of *B. napus*. The purple circle represents the distribution of DEGs on each chromosome. Light blue represents expression (log2 FC) of DEGs at 1 h vs 0 h, light green represents expression (log2 FC) of DEGs at 3 h vs 0 h and yellow represents expression (log2 FC) of DEGs at 24 h vs. 0 h.

**Figure 4 ijms-20-02771-f004:**
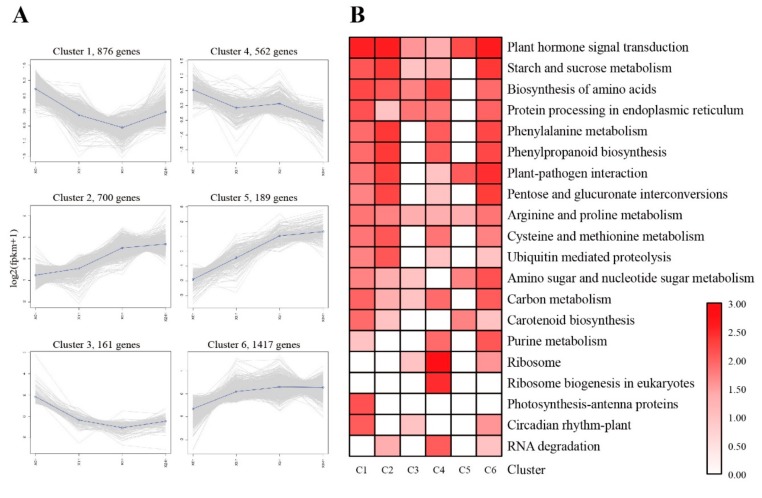
Co-expression clustering showing the expression profile of 3905 DEGs. (**A**) Six major clusters were identified along the time course of cold stress (0, 1, 3 and 24 h). The X-axis represents the time course of freezing stress (0, 1, 3 and 24 h). The Y-axis represents the value of the relative expression level (log2 (FPKM + 1). (**B**) Functional category enrichment among the six major clusters is based on KEGG annotation.

**Figure 5 ijms-20-02771-f005:**
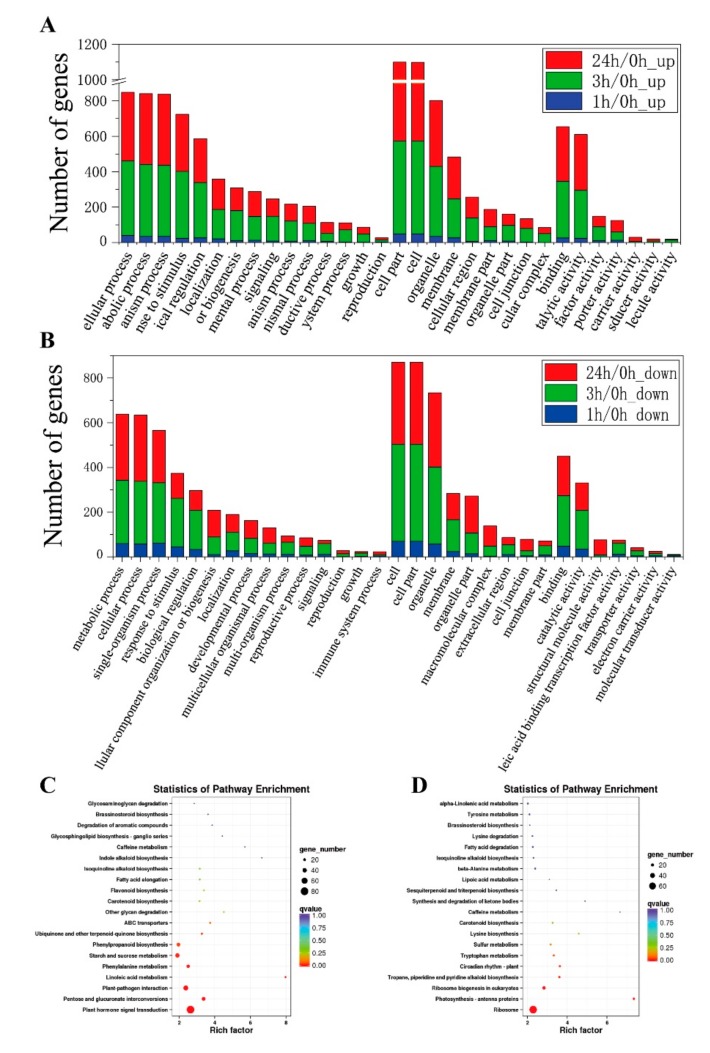
The functional annotation of differentially expressed genes (DEGs) in *Brassica napus* leaves under the freezing treatment. (**A**,**B**) are GO annotation, (**A**) up-regulated DEGs, (**B**) down-regulated DEGs. (**C**,**D**) are KEGG pathway enrichment, (**C**) up-regulated DEGs, (**D**) down-regulated DEGs.

**Figure 6 ijms-20-02771-f006:**
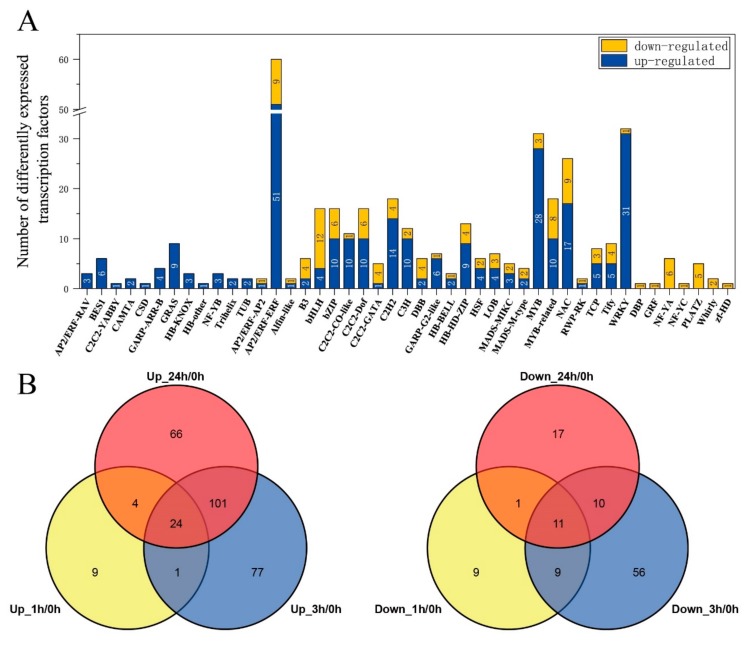
The distribution of differentially expressed transcription factors in *Brassica napus* leaves under the freezing treatment. (**A**) The histograms represent the number of up- or down-regulated transcription factors. (**B**) The Venn diagrams represent the distribution of transcription factors at different time points.

**Figure 7 ijms-20-02771-f007:**
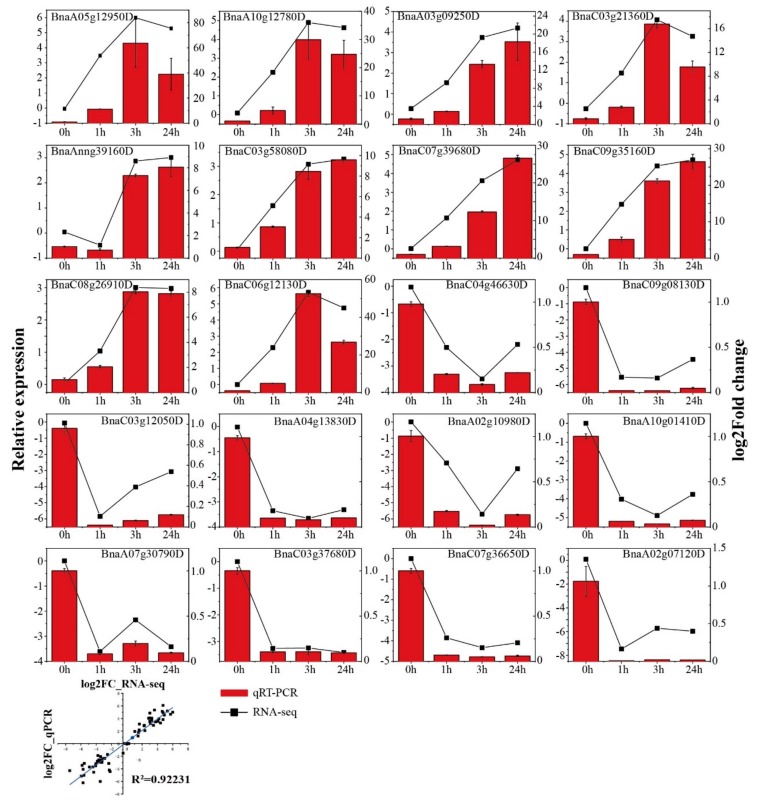
The qRT-PCR analysis of selected DEG genes in *Brassica napus* leaves under the freezing treatment. Error bars represent standard errors of the relative expression levels mean values by qRT-PCR (*n* = 4) (left y-axis). Broken lines represent transcript levels change (log2 FC) according to the FPKM value of RNA-Seq (right y-axis). Correlation between qRT-PCR and RNA-seq for select DEGs is also shown.

**Figure 8 ijms-20-02771-f008:**
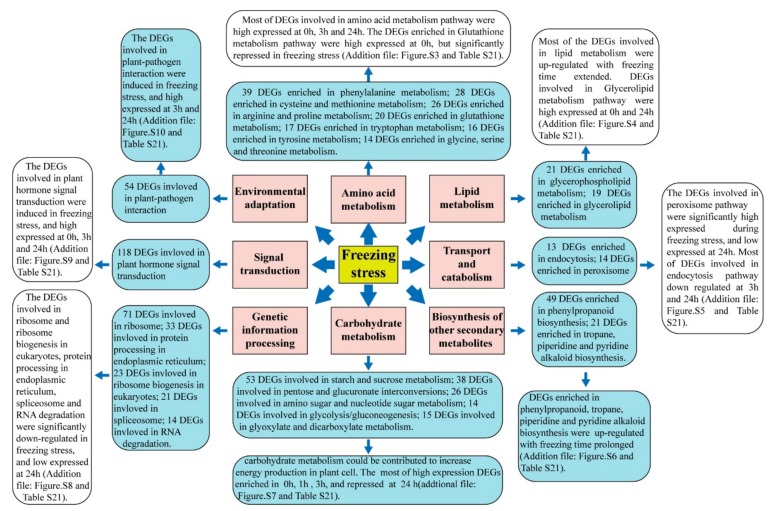
The differential responses of *Brassica napus* in response to freezing stress in transcriptome changes.

**Table 1 ijms-20-02771-t001:** The alignment statistics result with the reference gene for all samples.

Sample ID	Clean Reads	Clean Bases	GC Content	%≥Q30	Mapped Reads	Unique Match
0h	24,067,996	7,199,202,192	47.58%	92.87%	77.64%	71.46%
1h	23,250,784	6,954,053,329	47.65%	93.39%	77.38%	71.82%
3h	24,601,482	7,346,958,588	47.54%	93.39%	78.05%	72.47%
24h	25,630,541	7,664,124,316	47.41%	93.48%	78.05%	71.98%

clean reads: the number of clean reads, the single-ended meter; clean bases: the number of clean data; GC content: the percentage of GC-content in clean data; ≥Q30: Q-score of clean data ≥30; mapped reads: the number of reads mapped to the reference genome and its percentage in clean reads; unique mapped reads: the number of reads mapped to the only location of the reference genome and its percentage in clean reads; multiple mapped reads: the number of reads mapped to multiple locations of the reference genome and its percentage in clean reads.

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
