# Peer review of "Transcriptome Profile Analysis of Winter Rapeseed (Brassica napus L.) in Response to Freezing Stress, Reveal Potentially Connected Events to Freezing Stress"

_ijms, 2019, doi:10.3390/ijms20112771_

Reviewer 1 Report

The authors investigated trascriptome of winter rapeseed on several cold conditions and selected possible regulators and pathways based on GO and enrichment analyses. The amount of analyses and effort is enough to draft a single paper but the fundamental design is not impressive. Let me clarify one by one.

Major:

- Strain and data availability

The target strain, 2016TS(G)10, is probably not public and unavailable to other researchers. Please add information for the availability of this bioresource. In addition, trascriptome data should be available too. Please submit the data to a public repository such as GEO or GEA. Availability of data is a prerequisite of publication.

- Freezing treatment

Freezing stress of 24 h sounds long. Is this length common in previous research? Please describe the starting time of the freezing in the method section. Its timing relative to the photoperiod is important. Please discuss why this length was chosen.

- Genome analysis

There are not a few transcriptome papers on Brassica napus (including the most recent in Nature Communications 2019, 1154 Lu et al.). At least, some comparison or discussion is necessary in terms of world napus accessions. Please add information on the strain lineage.

From the mapping results, the distribution of DEGs in C subgenomes and A subgenomes should be clear. Also the characteristic of this particular strain needs to be explained in terms of cultivars.

Minor:

English needs a significant upgrade.

Author Response

Dear reviewer:

Thank you for your comments on our manuscript entitled “Transcriptome Profile Analysis of Winter Rapeseed (Brassica napus L.) in Response to Freezing Stress, Reveal Potentially Connected Events to Freezing Stress" (Manuscript ID: ijms-500621). Those comments are very helpful for revising and improving our paper, as well as the important guiding significance to other research. We have studied the comments carefully and made corrections which we hope meet with approval. The main corrections are in the manuscript and the responds to the reviewer’s comments are as follows.

Point 1: Strain and data availability

The target strain, 2016TS(G)10, is probably not public and unavailable to other researchers. Please add information for the availability of this bioresource. In addition, transcriptome data should be available too. Please submit the data to a public repository such as GEO or GEA. Availability of data is a prerequisite of publication.

Response 1: 2016TS(G)10 was formed by hybridization of Brassica rapa (longyou7 bred by Gansu Agricultural University) as female parent and Brasscia napus (Vision introduced from Europe) as male parent. The information of 2016TS(G)10 was added in our manuscript with the line number are 92-103.

The related sentences are as follows:

“2016TS(G)10 was formed by hybridization of Brassica rapa (longyou7, a strong cold tolerance variety, bred by Gansu Agricultural University) as female parent and Brassica napus (Vision, a winter ecotype, introduced from Europe) as male parent.”(Line 92-94)

The sequenced raw reads have been submitted to the SRA at NCBI (SRA accession: PRJNA540905). The related sentences are as follows: “The sequenced raw reads have been submitted to the SRA at NCBI with the accession numbers: SRP540905.” (Line560-561)

Point 2: Freezing treatment

Freezing stress of 24 h sounds long. Is this length common in previous research? Please describe the starting time of the freezing in the method section. Its timing relative to the photoperiod is important. Please discuss why this length was chosen.

Response 2: 2016TS(G)10, a strong winter ecotype, could overwinter in Lanzhou, northern China, without snow cover (overwintering rate is 25.5%). We want to explore the response of freezing tolerance in 2016TS(G)10 by short time and longtime freezing. We have chosen 1h, 3h and 24h in our studies. The relative previous researches are following:

 Triticum  aestivum L. exposed to -5°C treated for 3 days. “Han Q, Kang G, Guo T: Proteomic analysis of spring freeze-stress responsive proteins in leaves of bread wheat (Triticum aestivum L.). Plant physiology and biochemistry: PPB 2013, 63:236-244.”

 Loquat exposed to -3 °C treated for 24 h. “Xu HX, Li XY, Chen JW: Comparative transcriptome profiling of freezing stress responses in loquat (Eriobotrya japonica ) fruitlets. J Plant Res 2017, 130(5):1-15.”

Wheat exposed to -2°C for 24 h. “Si T, Wang X, Wu L, Zhao C, Zhang L, Huang M, Cai J, Zhou Q, Dai T, Zhu J-K et al: Nitric Oxide and Hydrogen Peroxide Mediate Wounding-Induced Freezing Tolerance through Modifications in Photosystem and Antioxidant System in Wheat. Front Plant Sci 2017, 8:1284-1284.”

Brassica rapa L. exposed to -4 ℃ for 8 h. “Zeng X, Xu Y, Jiang J, Zhang F, Ma L, Wu D, Wang Y, Sun W: iTRAQ-Based Comparative Proteomic Analysis of the Roots of TWO Winter Turnip Rapes (Brassica rapa L.) with Different Freezing-Tolerance. Int J Mol Sci 2018, 19(12).”

Alfalfa exposed to -8 °C for 3 h. “Shu Y, Li W, Zhao J, Zhang S, Xu H, Liu Y, Guo C: Transcriptome sequencing analysis of alfalfa reveals CBF genes potentially playing important roles in response to freezing stress. Genetics and molecular biology 2017, 40(4):824-833.”

Samples were collected from 8 AM, at which time point the non-treated samples (0 h) were obtained, and continued for 1 h (9 AM), 3 h (11 AM) and 24h (8 AM of the next day), respectively, to obtain frozen-treated samples.

The related sentences are as follows:

To avoid changes caused by the circadian rhythm, freezing stress treatments were started at 8 AM under light, which non-treated samples (0 h) were obtained, and continued for 1, 3 and 24h (8 AM of the next day), respectively, to obtain frozen-treated samples.” (Line 527-529).

Point 3: Genome analysis

There are not a few transcriptome papers on Brassica napus (including the most recent in Nature Communications 2019, 1154 Lu et al.). At least, some comparison or discussion is necessary in terms of world napus accessions. Please add information on the strain lineage.

From the mapping results, the distribution of DEGs in C subgenomes and A subgenomes should be clear. Also the characteristic of this particular strain needs to be explained in terms of cultivars.

Response 3: Thank you for your significant suggestion! According to your suggestion, we have carefully read recent publications about Brassica napus sequencing article, and quoted some of them in our manuscript.

The references are as follows:

11.           Wei L, Jian H, Lu K, Yin N, Wang J, Duan X, Li W, Liu L, Xu X, Wang R et al: Genetic and transcriptomic analyses of lignin- and lodging-related traits in Brassica napus. TAG Theoretical and applied genetics Theoretische und angewandte Genetik 2017, 130(9):1961-1973.

12.           Hossain Z, Pillai VS, Gruber MY, Yu M, Amyot L, Hannoufa A: Transcriptome profiling of Brassica napus stem sections in relation to differences in lignin content. Bmc Genomics 2018, 19(1):255.

13.           Lu K, Peng L, Zhang C, Lu J, Yang B, Xiao Z, Liang Y, Xu X, Qu C, Zhang K et al: Genome-Wide Association and Transcriptome Analyses Reveal Candidate Genes Underlying Yield-determining Traits in Brassica napus. Front Plant Sci 2017, 8:206.

14.           Sun C, Wang B, Yan L, Hu K, Liu S, Zhou Y, Guan C, Zhang Z, Li J, Zhang J et al: Genome-Wide Association Study Provides Insight into the Genetic Control of Plant Height in Rapeseed (Brassica napus L.). Front Plant Sci 2016, 7:1102.

15.           Gacek K, Bayer PE, Bartkowiak-Broda I, Szala L, Bocianowski J, Edwards D, Batley J: Genome-Wide Association Study of Genetic Control of Seed Fatty Acid Biosynthesis in Brassica napus. Front Plant Sci 2016, 7:2062.

16.           Jian H, Yang B, Zhang A, Ma J, Ding Y, Chen Z, Li J, Xu X, Liu L: Genome-Wide Identification of MicroRNAs in Response to Cadmium Stress in Oilseed Rape (Brassica napus L.) Using High-Throughput Sequencing. Int J Mol Sci 2018, 19(5).

17.           Zhang J, Mason AS, Wu J, Liu S, Zhang X, Luo T, Redden R, Batley J, Hu L, Yan G: Identification of Putative Candidate Genes for Water Stress Tolerance in Canola (Brassica napus). Front Plant Sci 2015, 6:1058.

18.           Wang P, Yang C, Chen H, Song C, Zhang X, Wang D: Transcriptomic basis for drought-resistance in Brassica napus L. Sci Rep 2017, 7:40532.

19.           Liu C, Zhang X, Zhang K, An H, Hu K, Wen J, Shen J, Ma C, Yi B, Tu J et al: Comparative Analysis of the Brassica napus Root and Leaf Transcript Profiling in Response to Drought Stress. Int J Mol Sci 2015, 16(8):18752-18777.

20.           Megha S, Basu U, Joshi RK, Kav NNV: Physiological studies and genome-wide microRNA profiling of cold-stressed Brassica napus. Plant physiology and biochemistry : PPB 2018, 132:1-17.

21.           Du C, Hu K, Xian S, Liu C, Fan J, Tu J, Fu T: Dynamic transcriptome analysis reveals AP2/ERF transcription factors responsible for cold stress in rapeseed (Brassica napus L.). Molecular Genetics & Genomics Mgg 2016, 291(3):1053-1067.

22.           Xiong AS, Jiang HH, Zhuang J, Peng RH, Jin XF, Zhu B, Wang F, Zhang J, Yao QH: Expression and function of a modified AP2/ERF transcription factor from Brassica napus enhances cold tolerance in transgenic Arabidopsis. Molecular biotechnology 2013, 53(2):198-206.

23.           Chen L, Zhong H, Ren F, Guo QQ, Hu XP, Li XB: A novel cold-regulated gene, COR25, of Brassica napus is involved in plant response and tolerance to cold stress. Plant Cell Rep 2011, 30(4):463-471.

24.           Lu K, Wei L, Li X, Wang Y, Wu J, Liu M, Zhang C, Chen Z, Xiao Z, Jian H et al: Whole-genome resequencing reveals Brassica napus origin and genetic loci involved in its improvement. Nat Commun 2019, 10(1):1154.

25.           Boulos C, France D, Shengyi L, Parkin IAP, Haibao T, Xiyin W, Julien C, Harry B, Chaobo T, Birgit S: Plant genetics. Early allopolyploid evolution in the post-Neolithic Brassica napus oilseed genome. Science 2014, 345(6199):950-953.

26.           Wu D, Liang Z, Yan T, Xu Y, Xuan L, Tang J, Zhou G, Lohwasser U, Hua S, Wang H et al: Whole-Genome Resequencing of a Worldwide Collection of Rapeseed Accessions Reveals the Genetic Basis of Ecotype Divergence. Molecular plant 2019, 12(1):30-43.

In China, winter Brassica napus is mainly distributed in the south of 35° N, altitude about 1100 m. Winter Brassica rapa (longyou7 is a strong winter freezing tolerance variety which overwinter rate more than 80.0% in North China [28-30]) is the only oilseed crop in northern China in winter. Sun et al. studied the feasibility of expanding winter Brassica rapa to the northwest and cold and arid regions of North China, and made breakthroughs in cold-resistant breeding. Planting area of winter rapeseed is rapid expansion to Xinjiang, Tibet, Heilongjiang, and the most northern regions of China, and generates significant economic and ecological benefits.

The references are as follows:

28.           Zeng X, Xu Y, Jiang J, Zhang F, Ma L, Wu D, Wang Y, Sun W: iTRAQ-Based Comparative Proteomic Analysis of the Roots of TWO Winter Turnip Rapes (Brassica rapa L.) with Different Freezing-Tolerance. Int J Mol Sci 2018, 19(12).

29.           Ma L, Coulter JA, Liu L, Zhao Y, Chang Y, Pu Y, Zeng X, Xu Y, Wu J, Fang Y et al: Transcriptome Analysis Reveals Key Cold-Stress-Responsive Genes in Winter Rapeseed (Brassica rapa L.). Int J Mol Sci 2019, 20(5).

30.           Zeng X, Xu Y, Jiang J, Zhang F, Ma L, Wu D, Wang Y, Sun W: Identification of cold stress responsive microRNAs in two winter turnip rape ( Brassica rapa L.) by high throughput sequencing. Bmc Plant Biol 2018, 18(1):52.

2016TS(G)10, a strong winter ecotype, was formed by hybridization of Brassica rapa (longyou7) and Brassica napus (Vision). According to our previous studies, “2016TS(G)10” could overwinter in Lanzhou (Gansu, China, 36°7′ N, altitude 1,517 m), which is cold and arid region of northwest China. The overwintering rates for two consecutive years were 25.5% (non-mulch) ~80.0% (plastic mulch). Before overwintering (samples were harvested on 26 October), LT50 of field plant leaves was -13.38 (article submitted in Scientia Agricultura Sinica, under review).

About the distribution of DEGs in subgenomes C and subgenomes A, we add some related sentences in our manuscript, the details are as follows:

Result:

“To be clear the distribution of DEGs in subgenomes C and subgenomes A, we had drawn a Circos plot to show the distribution of DEGs on 19 chromosomes. 3,109 DEGs were accurately positioning on 19 chromosomes, 1,690 DEGs on subgenomes C and 1,419 on subgenomes A.” (Line 207-211).

Circos Plot added line 212.

Explanatory: “(D) Circos Plot show that the distribution of DEGs on 19 chromosomes and expression of DEGs in different time point. Red and blue showed the sizes of the 19 chromosomes of B. napus. The purple circle represents distribution of DEGs on each chromosome. Light blue represents expression (log2 FC) of DEGs at 1 h vs 0 h, light green represents expression (log2 FC) of DEGs at 3 h vs 0 h and yellow represents expression (log2 FC) of DEGs at 24 h vs 0 h.” (Line 216-220).

Method

 “According to the information resource database of B. napus (http: //www.genoscope.cns.fr.), the positional information of all DEGs was investigated. The positions of DEGs on chromosomes were drawn using Circos v0.69 [89].” (Line 572-574).

More details please see revised manuscript.

Minor:

English needs a significant upgrade.

Response: our manuscript has been polished by Dr. Wenyun Shen and Dr. Sun Jia, come from Saskatoon, Canada.

Reviewer 2 Report

There are two major flaws in this manuscript, one is related to the design of the experiment and the second is related to the thoroughness of the analysis.

The experiment design is based on comparing the cold-treated plants to the untreated plants. But, the untreated plants are not from the same time point as the cold-treated ones. So, when the authors compare gene expression between cold-stressed and non-stressed plants, the gene expression changes are caused by both the cold treatment as well as the circadian and diurnal cycles. This problem is particularly serious in the 3h/0h comparison since a large body of research suggests that at least one-third of all genes are under circadian control. It's unacceptable to include the 3h/0h comparison since no attention was paid to the circadian regulation of genes. If the authors want to include this time point they should repeat the experiment with the stressed and non-stressed plants grown in parallel at -2C and 24C and compare, using qPCR, both these samples to the 0h samples. This will show that the changes they see in the 3h/0h analysis are due to cold and not due to circadian regulation.

The authors measured cold-induced changes in plant morphology and physiology, and observed clear differences in soluble protein, sugar and Proline levels. Yet, when they generate the list of DE genes and conduct a pathway analysis they do not try to link this data to the observed physiological changes. For example, since the proline levels are clearly increased (Fig 1C) which of the genes in the proline metabolism pathway are correlated with this change? Does the expression of plastidial genes change?

Overall, this manuscript spends a lot of effort in describing the various genome-level analysis they have done, but do not provide any insights into the effect of cold stress on specific genes or processes. The only mention the authors make of the known cold-responsive pathway is to state that the CBFs do not respond in this experiment. While clearly important, CBFs are not the only known regulators of cold response. Multiple articles and reviews of cold response mechanisms in Arabidopsis have described 10-15 TFs that respond to cold independent of the CBF pathway. It is important that the authors describe how these alternate pathways respond to cold in Brassica napus.

Author Response

Dear reviewer:

Thank you for your comments on our manuscript entitled “Transcriptome Profile Analysis of Winter Rapeseed (Brassica napus L.) in Response to Freezing Stress, Reveal Potentially Connected Events to Freezing Stress" (Manuscript ID: ijms-500621). Those comments are very helpful for revising and improving our paper, as well as the important guiding significance to other research. We have studied the comments carefully and made corrections which we hope meet with approval. The main corrections are in the manuscript and the responds to the reviewers’ comments are as follows.

Point 1The experiment design is based on comparing the cold-treated plants to the untreated plants. But, the untreated plants are not from the same time point as the cold-treated ones. So, when the authors compare gene expression between cold-stressed and non-stressed plants, the gene expression changes are caused by both the cold treatment as well as the circadian and diurnal cycles. This problem is particularly serious in the 3h/0h comparison since a large body of research suggests that at least one-third of all genes are under circadian control. It's unacceptable to include the 3h/0h comparison since no attention was paid to the circadian regulation of genes. If the authors want to include this time point they should repeat the experiment with the stressed and non-stressed plants grown in parallel at -2C and 24C and compare, using qPCR, both these samples to the 0h samples. This will show that the changes they see in the 3h/0h analysis are due to cold and not due to circadian regulation.

Response 1Samples were collected from 8 AM, at which time point the non-treated samples (0 h) were obtained, and continued for 1 h (9 AM), 3 h (11 AM) and 24h (8 AM of the next day), respectively, to obtain frozen-treated samples.

The related sentences are as follows:

To avoid changes caused by the circadian rhythm, freezing stress treatments were started at 8 AM under light, which non-treated samples (0 h) were obtained, and continued for 1, 3 and 24h (8 AM of the next day), respectively, to obtain frozen-treated samples.” (Line 527-529).

Thank you for your significant suggestion! According to your suggestion, we have been preparing plant materials, and repeat the experiment with the freezing-stressed (-2) and non-stressed (22℃) plants grown in parallel, and compare, using qPCR, both these samples to the 0 h samples. We selected 9 DE genes with high FPKM and fold change. The results of qPCR have shown that at room temperature, 6 up-regulated DE genes did not change significantly with the extension of time, but these genes notably changed under freezing stress. Meanwhile, 2 out of 3 down-regulated DE genes were markedly down regulated in response to freezing stress, but also have some extent change at room temperature with the prolonging of time (Figure S11).

BnaC03g12050D (Non-specific lipid-transfer protein 3) and BnaA02g07120D (Non-specific lipid-transfer protein 4) belong to LTP family members. In Arabidopsis thaliana ecotype Col-0 plant, LTP3 was able to bind to lipids, and expressed ubiquitously, meanwhile, the LTP3 protein was localized to the cytoplasm and cell membrane [1, 3]. According to cis-acting motifs in its promoter sequences of LTP3, it was deduced to be regulated by cold, ABA, MeJA, Auxin and oxidative stress [3].The research shows that LTP3 was involved in Arabidopsis plants tolerance to freezing, drought stress and immunity [2,4], also induced by prolonged water deficit, salt, ABA treatment and heat stress [3]. LTP3 acts as a target of MYB96 to be involved in plant tolerance to freezing and drought stress. LTP3 was positively regulated by MYB96 via the direct binding to the LTP3 promoter. Overexpression of LTP3 resulted in constitutively enhanced freezing tolerance without affecting the expression of CBFs and their target COR genes [2]. Overexpression of TaLTP3 in yeast and Arabidopsis plants enhanced their heat tolerance. The TaLTP3 confers heat stress tolerance possibly through reactive oxygen species (ROS) scavenging [3]. LTP3 is a novel negative regulator of plant immunity which contributes to disease susceptibility in Arabidopsis by the manipulation of the ABA-SA balance [4]. Arabidopsis ltp3 mutant is compromised in germination and seedling growth [5]. In our study, 16 DEGs belong to LTP family, 9 DEGs down-regulated with high FPKM and log2FC, and 7 DEGs up-regulated, which is the FPKM and log2FC more lower than up-regulation ones. Why does the expression of these two genes (BnaC03g12050D and BnaA02g07120D) change with the prolonging of time at room temperature, we require further experiments and analysis.

1.    Kader JC: LIPID-TRANSFER PROTEINS IN PLANTS. Annu Rev Plant Physiol Plant Mol Biol 1996, 47:627-654.

2.   Guo L, Yang H, Zhang X, Yang S: Lipid transfer protein 3 as a target of MYB96 mediates freezing and drought stress in Arabidopsis. J Exp Bot 2013, 64(6):1755-1767.

3.   Wang F, Zang XS, Kabir MR, Liu KL, Liu ZS, Ni ZF, Yao YY, Hu ZR, Sun QX, Peng HR: A wheat lipid transfer protein 3 could enhance the basal thermotolerance and oxidative stress resistance of Arabidopsis. Gene 2014, 550(1):18-26.

4.   Gao S, Guo W, Feng W, Liu L, Song X, Chen J, Hou W, Zhu H, Tang S, Hu J: LTP3 contributes to disease susceptibility in Arabidopsis by enhancing abscisic acid (ABA) biosynthesis. Molecular plant pathology 2016, 17(3):412-426.

5.   Pagnussat LA, Oyarburo N, Cimmino C, Pinedo ML, de la Canal L: On the role of a Lipid-Transfer Protein. Arabidopsis ltp3 mutant is compromised in germination and seedling growth. Plant signaling & behavior 2015, 10(12):e1105417.

Point 2The authors measured cold-induced changes in plant morphology and physiology, and observed clear differences in soluble protein, sugar and Proline levels. Yet, when they generate the list of DE genes and conduct a pathway analysis they do not try to link this data to the observed physiological changes. For example, since the proline levels are clearly increased (Fig.1 C) which of the genes in the proline metabolism pathway are correlated with this change? Does the expression of plastidial genes change?

Response 2We carefully analyzed the DE genes related with the physiological changes. The relevant supplementary contents have been added in our manuscript.

The related sentences are as follows:

“Proline is well documented to be associated with plant abiotic tolerance in multiple ways … … In this study, the expression profile of… …might play some essential roles in response to freezing stress of Brassica napus.” (Line 383- 403)

Point 3Overall, this manuscript spends a lot of effort in describing the various genome-level analyses they have done, but do not provide any insights into the effect of cold stress on specific genes or processes. The only mention the authors make of the known cold-responsive pathway is to state that the CBFs do not respond in this experiment. While clearly important, CBFs are not the only known regulators of cold response. Multiple articles and reviews of cold response mechanisms in Arabidopsis have described 10-15 TFs that respond to cold independent of the CBF pathway. It is important that the authors describe how these alternate pathways respond to cold in Brassica napus.

Response 3We read related reviews and articles, then, analyzed DE transcription factor. The relevant supplementary contents have been added in our manuscript.

The related sentences are as follows:

Although the CBFs play important roles in cold acclimation … …that are co-expressed in parallel with CBF2 … …could be the targets for further functional characterization.” (Line 489-511)

More details please see revised manuscript.

Round  2

Reviewer 1 Report

Authors responded sincerely and the manuscript became much better now.

Reviewer 2 Report

Thank you for the revisions. The manuscript is much improved with the additions and changes you made.